# Serum Progranulin Level Might Differentiate Non-IPF ILD from IPF

**DOI:** 10.3390/ijms24119178

**Published:** 2023-05-24

**Authors:** Nóra Melinda Tóth, Veronika Müller, Tamás Nagy, Lőrinc Polivka, Péter Horváth, Anikó Bohács, Noémi Eszes

**Affiliations:** Department of Pulmonology, Semmelweis University, 1083 Budapest, Hungary; toth.nora1@med.semmelweis-univ.hu (N.M.T.); muller.veronika@semmelweis.hu (V.M.); polivka.lorinc@semmelweis.hu (L.P.); horvath.peter2@med.semmelweis-univ.hu (P.H.); bohacs.aniko@med.semmelweis-univ.hu (A.B.)

**Keywords:** idiopathic pulmonary fibrosis, progranulin, interstitial lung disease, usual interstitial pneumonia, differential diagnosis

## Abstract

Diagnosing interstitial lung disease (ILD) can be a challenging process. New biomarkers may support diagnostic decisions. Elevated serum progranulin (PGRN) levels have been reported in liver fibrosis and dermatomyositis-associated acute interstitial pneumonia. Our aim was to assess the role of PGRN in the differential diagnosis of idiopathic pulmonary fibrosis (IPF) and other ILDs. Serum levels of PGRN were measured by enzyme-linked immunosorbent assay in stable IPF (*n* = 40), non-IPF ILD (*n* = 48) and healthy controls (*n* = 17). Patient characteristics, lung function, CO diffusion (DLCO), arterial blood gases, 6-min walk test, laboratory parameters and high-resolution (HR)CT pattern were assessed. In stable IPF, PGRN levels did not differ from healthy controls; however, serum PGRN levels were significantly higher in non-IPF ILD patients compared to healthy subjects and IPF (53.47 ± 15.38 vs. 40.99 ± 5.33 vs. 44.66 ± 7.77 ng/mL respectively; *p* < 0.01). The HRCT pattern of usual interstitial pneumonia (UIP) was associated with normal PGRN level, while for non-UIP patterns, significantly elevated PGRN level was measured. Elevated serum PGRN levels may be associated with non-IPF ILD, especially non-UIP patterns and might be helpful in cases of unclear radiological patterns in the differentiation between IPF and other ILDs.

## 1. Introduction

Interstitial lung diseases (ILD) are a large heterogeneous group of lung diseases, including more than 150 entities that have many clinical and pathophysiological features in common but differ significantly in etiology, therapy and prognosis [1].

Idiopathic pulmonary fibrosis (IPF) is the most common form of ILDs, with a progressive course and with an almost invariably poor prognosis [2]. It mainly manifests with the radiological and histological pattern of usual interstitial pneumonia (UIP) [3,4]. Connective tissue diseases (CTD) are often associated with ILD and should always be excluded before diagnosing IPF or other idiopathic interstitial pneumonia (IIP) [3,4]. Despite the wide range of diagnostic tools available today, differential diagnosis of ILDs can still be a major challenge to clinicians, and the gold standard for diagnosis should be established via multidisciplinary discussion (MDD) by a team of pulmonologists, radiologists and pathologists. In most ILD patients, it is possible to make a consensus diagnosis, while in many cases overlapping radiological and pathological patterns, clinical features, and incomplete data (e.g., lung biopsy is not feasible due to the patient’s advanced condition) may cause diagnostic uncertainty [5]. However, establishing an accurate diagnosis as soon as possible would be critical in initiating appropriate treatment and establishing the expected prognosis. Although a number of potentially significant biomarkers have been described in recent years [6], none have been introduced widely into clinical practice to date.

Progranulin (PGRN) is a 593 amino acids long, highly glycosylated secretory protein that is a precursor molecule to granulins that modulate cell growth and proliferation [7]. PGRN itself plays a role as a growth factor in many physiological and pathological processes, including inflammation, tissue regeneration and wound healing. PGRN mRNA is highly expressed in epithelial cell lines [8,9]. Elevated serum PGRN levels have been described in a variety of CTDs, such as systemic lupus erythematosus (SLE) [10,11,12], polymyositis (PM) and dermatomyositis (DM) [13]. Serum PGRN levels were measured in DM-associated ILD, and elevated levels were found in acute/subacute interstitial pneumonia compared to both DM patients without ILD and those with chronic interstitial pneumonia [13]. These data may also suggest a role for PGRN in the development of ILD, especially with underlying autoimmunity.

Our aim was to examine the role of PGRN in ILDs with different etiologies and to assess its significance as a potential serum biomarker in the differential diagnosis of ILDs of possible autoimmune background.

## 2. Results

The study participants’ characteristics are shown in Table 1. IPF patients were significantly older and, more frequently, men. Laboratory parameters were similar in all groups, except for autoimmune serology, as this was more common in patients with CTD-ILD.

Lung function parameters confirmed mild restrictive ventilator patterns in IPF patients (Table 2). Significantly lower D_LCO_ and K_LCO_ values were confirmed in IPF. Non-IPF ILD was associated with higher pO_2_ and lower pCO_2_ values. No difference between groups in functional performance assessed by 6MWT was present. The HRCT patterns did significantly differ between groups, UIP/pUIP being the main pattern in IPF, while NSIP in CTD-ILD and other patterns in the other ILD subgroups.

The PGRN serum level in healthy controls was 40.99 ± 5.33 ng/mL, and the upper limit of normal (ULN) was set at mean ± 2SD representing the value: 51.65 ng/mL. All patients’ PGRN level was significantly higher as compared to controls. The IPF patients had similar PGRN levels as the healthy controls, while significantly lower as in non-IPF ILD patients (Figure 1).

When CTD-ILD and other ILD subgroups were additionally analyzed, they both showed significantly higher values than controls, but only the other ILD subgroup differed from IPF patients (Figure 2).

Patients above ULN were more often patients with non-UIP HRCT morphology, and in the presence of UIP patterns, normal PGRN levels were more common (Figure 3).

No correlation was found between PGRN level and lung function, blood gases, 6MWT parameters, GAP stage or laboratory parameters (CRP level, eGFR, autoimmune serology). Five-step multivariate linear regression analysis showed that higher age and male sex predicts higher PGRN levels and confirmed that IPF diagnosis is associated with lower PRGN levels than non-IPF; however, based on step 5, we can assume that the UIP pattern is more strongly associated with lower PGRN levels than an IPF diagnosis. Immunosuppressive and antifibrotic treatment did not influence the PGRN values (Appendix A).

## 3. Discussion

This cross-sectional study was the first to investigate the serum levels of PGRN in patients with stable IPF and other non-IPF ILDs, including CTD-ILD. PGRN, as a pleiotropic growth factor, has been shown to mediate immune response and tissue regeneration in addition to many other biological processes. It was shown to be involved in the pathogenesis of liver fibrosis [14], acute ILD associated with DM [13] and its role has been even raised in acute exacerbation of IPF according to a small single-center study [15], but its potential function in other ILDs has not yet been examined.

The PGRN concentrations determined in our study were similar to those measured by Tanaka et al. and Xie et al. in DM and in IPF [13,15].

The age distribution of our patient population is in line with previous studies and clinical practice. Patients with IPF were older than non-IPF ILD patients; this difference was even more pronounced compared to the CTD-ILD subgroup. The mean age of 68.5 years in IPF patients is consistent with literature data and our former studies on IPF patients [16,17,18]. The age of healthy controls was similar to that of non-IPF ILD patients. However, there was no correlation between serum PGRN levels and age.

According to previous studies, the proportion of men with IPF was higher as compared to non-IPF ILD patients [16,17,18]. A significant proportion of the non-IPF ILD group consisted of patients with autoimmune diseases characterized by female predominance, resulting in a higher number of women in this group. It is important to note that serum PRGN was independent of the patient’s sex. Decreased renal function [19] and the presence of diabetes [20,21] may affect PGRN levels. In our study, there was no difference in renal function and proportion of diabetic patients between the groups.

IPF patients were characterized by mild restrictive ventilation disorder. Although the majority of patients with IPF had a mild to moderate disease according to the GAP index, the non-IPF ILD group performed significantly better in terms of CO diffusion parameters than IPF patients and the pO_2_ was also higher in this population. This may be due to the fact that the majority of non-IPF ILD patients had limited lung involvement, which in the vast majority of cases was below 20%. Overall, the functional condition of the examined patients was stable at the time of the study, with good 6MWT performance in all subjects.

PGRN concentrations were significantly higher in patients with non-IPF ILD compared to patients with IPF, while the PGRN levels of the IPF group were comparable to those in healthy controls. The similarity between the IPF patients and the healthy group may be explained by the fact that all patients were stable, and the majority received antifibrotic therapy. On the other hand, previous research suggests a stronger role for PGRN in acute and rapidly progressive ILDs. Examining ILD associated with DM, significantly higher serum PGRN concentrations were measured in patients with acute and subacute interstitial pneumonia than in those with DM with chronic interstitial pneumonia [13]. In this previous study, PGRN levels correlated with activity markers (ferritin, lactate dehydrogenase, CRP) associated with pulmonary cell damage caused by the autoimmune inflammatory process, but they didn’t correlate with Krebs von den Lungen-6, which rather reflects cell proliferation and regeneration associated with ILD. Moreover, PGRN’s role in wound healing has been described in fibroblast accumulation during early granulation but not in the subsequent collagen deposition phase [22]. It is also known that PGRN is the co-factor for toll-like receptor 9 (TLR9) [23], and the expression of TLR9 in biopsies of fast-progressing IPFs was found to be significantly higher than in slow-progressing patients [24]. Besides, the TLR9 agonist cytosine-phosphorothioate-guanine (CpG) oligonucleotide was able to induce acute exacerbation in mouse lung transplanted with fibroblasts of rapid progressors [24]. The exact association of TLR9 with serum levels of PGRN remains to be elucidated; however, in a recent study, significantly higher levels of PGRN were indeed described in acute exacerbation of IPF compared to stable disease [15]. After all, based on the data so far, it can be assumed that the chronic, stable disease in IPF is characterized by a lower PGRN value.

The PGRN levels were higher in the non-IPF patients than in the healthy group, which may indicate a possible pathogenetic role of PGRN in this population. For a more accurate understanding of these results, an individual study of each ILD would be required. We distinguished patients with CTD-ILDs from other non-IPF ILDs because serum PGRN has been described as a promising biomarker in DM-associated ILD in a previous study [13]. Moreover, nearly half of our non-IPF patient population consisted of CTD-ILD patients.

Elevated PGRN levels in CTD-ILD compared to healthy controls may be explained by the presence of systemic autoimmune disease itself, as several of PGRN’s functions are known to play a role in autoimmune processes [25]. This might be underlined by previous data showing that PGRN concentrations were elevated in DM without the presence of ILD being noted [13]. Furthermore, several studies have shown elevated serum PGRN levels in SLE [10,11,12] that correlated with disease activity [11], and successful management of active SLE decreased the PGRN levels [11]. Our observation that after the subdivision of the non-IPF group, the statistically significant difference disappeared between IPF and CTD-ILD may be explained by the small number of elements in the groups, which is a limitation of our study. Moreover, in light of the current literature [13,15], the assessment of PGRN’s role in the differential diagnosis in our study is complicated by the fact that mainly stable, treated patients were included in all groups.

In the other ILD subgroup, PGRN levels were also elevated compared to healthy subjects. As this group was quite heterogeneous (sarcoidosis, HP, COP, DI-ILD and LAM), none of the diseases represented themselves with an evaluable number of elements; therefore, conclusions specific to each ILD cannot be drawn. However, the pathogenesis of these diseases, similarly to CTD-ILDs, is driven by inflammatory mechanisms and consequent activation of the immune system. Since PGRN is reported to take an important part in the inflammatory response [22], in this case, it may be a marker of the ongoing inflammatory and immune-mediated processes.

There are potential factors such as IL-6 [11,26] and tumor necrosis factor α [26] that can influence serum PGRN levels; however, in this study, these cytokine levels were not measured. There are limited data available about the potential influence of immunosuppressants, especially steroids, on the level of PGRN [26]. Our results showed that neither immunosuppressive nor antifibrotic treatment was associated with the PGRN levels. Our linear regression model also indicated that higher age and male sex predicts higher levels of PGRN. Although the IPF group was significantly older and more prominently male, linear regression still showed a strong association between IPF and lower PGRN levels.

The elevated PGRN levels seen in non-IPF ILD compared to the IPF group may provide a new opportunity for the differential diagnosis of IPF from other ILDs. The difference can be explained by the different pathogenesis of the diseases. IPF is a chronic, relentlessly irreversible disease where epithelial cell activation and aberrant epithelial cell repair predominate over inflammatory processes. However, the level of PGRN may rise in acute exacerbation, indicating the altered pathomechanism, as mentioned above.

Although the UIP HRCT pattern is characteristic of IPF, non-IPF ILDs, including CTD-ILDs, can also present with UIP; therefore, the differentiation is often difficult not only clinically but also radiologically [4]. The association of non-UIP pattern with a PGRN level above ULN and a UIP pattern with normal PGRN levels can help the decision-making in cases of unclear radiological diagnoses.

A limitation of our study is the low numbers in the specific ILD subgroups, the cross-sectional design, the homogenous Caucasian population and the possible treatment effect of antifibrotics in the IPF group.

In summary, serum PGRN level may be a useful biomarker in the differential diagnosis of IPF and non-IPF ILDs, especially in cases where a clear decision cannot be made due to an uncertain HRCT pattern and sufficient clinical information or histological confirmation is not available.

## 4. Materials and Methods

### 4.1. Study Population

We performed a cross-sectional study of serum PGRN levels in 88 patients diagnosed with ILD and 17 healthy controls at the Department of Pulmonology Semmelweis University between October 2016 and October 2018. Consecutive patients presenting at the ILD care unit, giving their consent for the analysis, were enrolled. Each ILD diagnosis was made by the ILD MDD team at Semmelweis University. Our MDD team consists of 3 pulmonologists, expert radiologists, clinical immunologists with expertise in ILD and a pathologist if histology was present, according to international guidelines [27,28]. All patients were diagnosed with ILD within 2 months to 3 years prior to the sampling, and treated patients were on stable dose antifibrotic or immunosuppressive medication for at least 4 weeks at the time of the testing. At the time of sampling, detailed medical histories were taken, and physical examinations, respiratory function tests, arterialized capillary blood gas analysis (ABG) and laboratory tests, including C-reactive protein (CRP) and estimated glomerular filtration rate (eGFR), were performed in all patients. Demographic data, high-resolution computed tomography (HRCT) patterns, 6-min walk test (6MWT) parameters and autoantibody profiles were collected from medical records. Gender-age-physiology (GAP) index used for clinical severity prediction in IPF was assessed in all patients [29]. The control group consisted of healthy staff members of the Department of Pulmonology. All participants were of Caucasian origin, representing the Hungarian population.

### 4.2. Pulmonary Evaluation, Functional Measurements and Radiological Patterns

Analyses of forced vital capacity (FVC), forced expiratory volume in 1 s (FEV_1_), FEV_1_/FVC and total lung capacity (TLC) were performed by electronic spirometer according to the current guidelines at the Department of Pulmonology as previously described [30]. Lung diffusion capacity was measured using the single-breath CO method (D_LCO_), and the transfer coefficient of the lung for CO (K_LCO_) was also calculated (PDD: 301/s, Piston, Budapest, Hungary). Exercise capacity was determined using the 6MWT: 6-min walk distance (6MWD) in meters, baseline and post-exercise oxygen saturation (SpO_2_), heart rate and the Borg scale referring to dyspnea were assessed. ABG was measured from arterialized capillary blood (Cobas b 221, Roche, Mannheim, Germany).

HRCT scan was performed in all patients: inspiration and expiration positions (Ingenuity Core 64 and Brilliance 16 CT scanners, Philips, Amsterdam, The Netherlands), the pattern of the ILD was determined by radiology experts.

### 4.3. Classification of ILD Patients

The ILD population was divided into 2 main groups as being IPF (*n* = 40) and non-IPF ILD (*n* = 48) patients. The non-IPF ILD group was separated into 2 subgroups: CTD-ILD (*n* = 20) and other ILD (*n* = 28). CTDs included systemic sclerosis (*n* = 8), idiopathic inflammatory myopathies including PM/DM (*n* = 5), SLE (*n* = 2), rheumatoid arthritis (*n* = 1), Sjögren’s syndrome (*n* = 1), mixed connective tissue disease (*n* = 1) and overlap cases (*n* = 2). Other ILD group consisted of subjects with sarcoidosis (*n* = 9), hypersensitive pneumonitis (HP, *n* = 6), drug-induced ILD (DI-ILD, *n* = 4), granulomatosis with polyangiitis (GPA, *n* = 1), cryptogenic organizing pneumonia (COP, *n* = 2), idiopathic nonspecific interstitial pneumonia (NSIP, *n* = 1), acute interstitial pneumonia (AIP, *n* =1), lymphangioleiomyomatosis (LAM, *n* = 1) and other rare forms of ILD (*n* = 3).

### 4.4. Measurement of Serum PGRN Levels

Blood samples were kept at room temperature until clot formation (30 min), then centrifuged at 1800× *g*. The separated 1 mL serum samples were kept frozen at −80 °C until measurement. Serum PGRN levels were determined by using the Quantikine^®^ Human Progranulin immunoassay (R&D Systems, Inc., Minneapolis, MN, USA)—a quantitative sandwich enzyme-linked immunosorbent assay (ELISA)—according to the user manual.

### 4.5. Ethical Statement

The study was conducted in accordance with the Declaration of Helsinki and approved by the Ethical Committees of Semmelweis University (Study No. 16/2018) and ETT TUKEB (Study No. 69/2015). Written informed consent was obtained from all subjects prior to the sampling.

### 4.6. Statistical Analysis

GraphPad Prism 9^®^ (GraphPad Software, Inc., San Diego, CA, USA) and IBM SPSS^®^ Statistics 28.0 (IBM Corp., Armonk, NY, USA) were used for statistical analysis. Data were expressed as mean ± SD for the Gaussian distribution and as median (interquartile range) for non-normally distributed variables. The normality of the data distribution was determined by the D’Agostino-Pearson test. The clinical parameters of the groups were compared by *t*-test or Mann-Whitney test, ANOVA and Chi^2^ test. Welch’s ANOVA was used for multiple comparisons of PGRN levels between the groups since the studied populations’ data had unequal variances. The relationship between PGRN levels and functional parameters was analyzed with Pearson’s correlation analysis in the case of normally distributed data. Spearman’s rank-order correlation was used when a nonparametric test was needed. After performing the univariate tests, in order to assess the role of possible confounding factors on the PGRN levels, we performed a stepwise multivariate linear regression analysis on the univariate data that proved to be significant according to the following steps: Step 1. Age and sex; Step 2. Medication included; Step 3. The presence of ILD included; Step 4. IPF or non-IPF diagnosis included; Step 5. HRCT pattern included (Appendix A). *p* < 0.05 was considered statistically significant.

## 5. Conclusions

Differentiating IPF from non-IPF ILDs can be a difficult task due to similar clinical features and often unclear radiological patterns, especially when invasive tests cannot be performed to support the diagnosis histologically. Measuring serum PGRN levels may help the decision-making in such uncertain cases, as our results showed that elevated progranulin levels are associated with non-IPF ILD and non-UIP HRCT patterns. It may be worthwhile to investigate the role of PGRN in different non-IPF ILDs in larger studies. Future research on the association between serum PGRN and the activity of ILDs and the therapeutic response would be warranted for a better understanding of the multifaceted functions of this protein in the pathomechanism of ILDs.

## Figures and Tables

**Figure 1 ijms-24-09178-f001:**
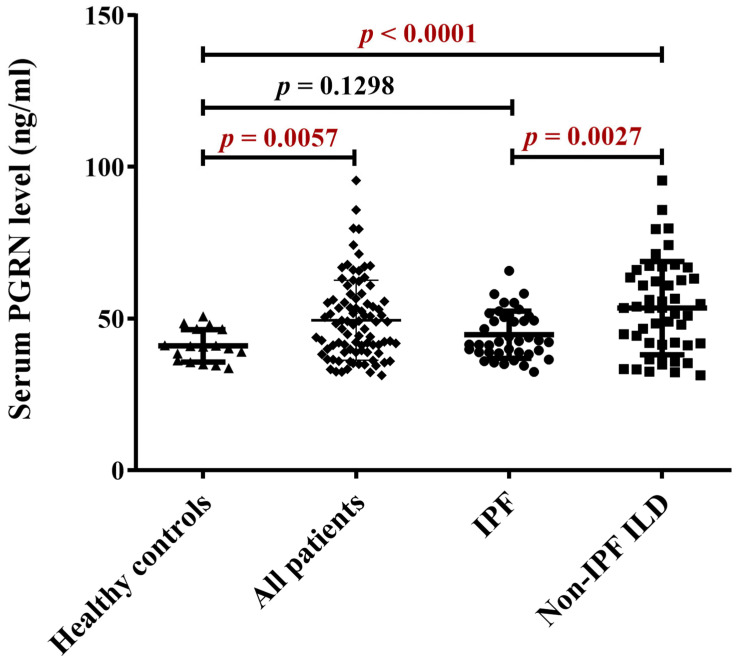
Comparison of serum PGRN levels among healthy, all patients, IPF and non-IPF-ILD groups. The PGRN level of all patients is significantly increased as compared to healthy controls. IPF and the healthy groups are similar. Serum PGRN is higher in the non-IPF ILD group compared to the stable IPF patients or healthy controls.

**Figure 2 ijms-24-09178-f002:**
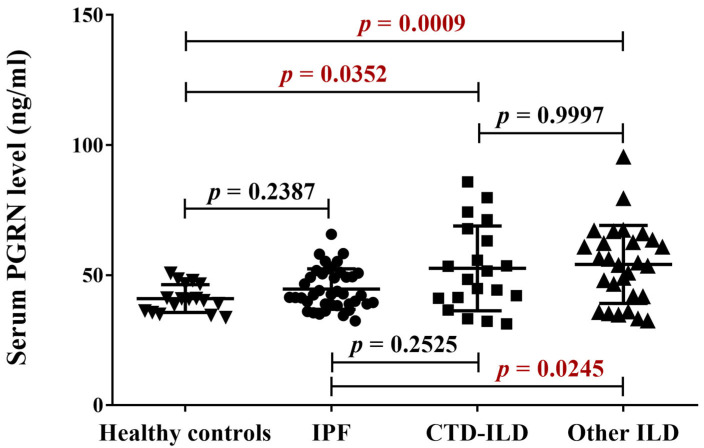
Comparison of serum PGRN levels among the healthy, IPF, CTD-ILD and other ILD groups. PGRN levels of IPF and the healthy groups are similar. Serum PGRN is higher in the CTD-ILD and the other ILD group compared to the healthy controls. The other ILD group’s PGRN level is significantly higher than that of the IPF patients.

**Figure 3 ijms-24-09178-f003:**
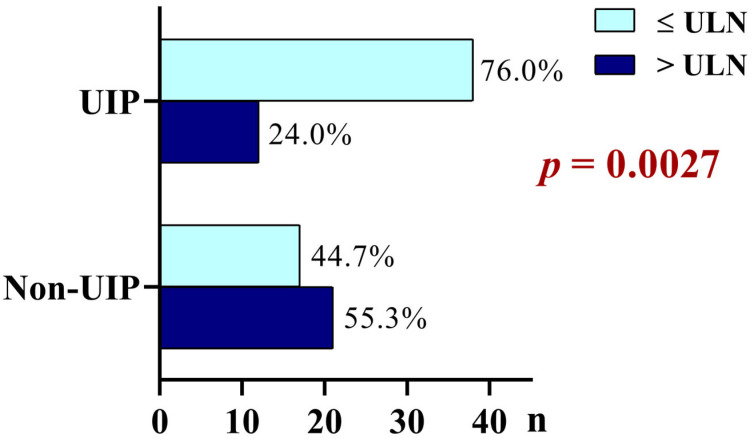
Frequencies of serum PGRN levels ≤ and > upper limit of normal (ULN; 51.65 ng/mL) in patients with UIP and non-UIP HRCT patterns. HRCT pattern is associated with PGRN levels (Pearson Chi-Square Test, *p* = 0.0027). In patients with non-UIP patterns, the PGRN level is more > ULN, while in the presence of UIP patterns, normal PGRN levels are more common.

**Table 1 ijms-24-09178-t001:** Patient characteristics.

Parameters	Healthy Controls (*n* = 17)	All ILD Patients (*n* = 88)	IPF (*n* = 40)	Non-IPF ILD (*n* = 48)	CTD-ILD (*n* = 20)	Other ILD (*n* = 28)
Age (year)	53.12 ± 13.49	64.00 (57.25–71.00)	**68.58 ± 6.47 ***	**58.46 ± 13.74 ^#^**	**55.10 ± 15.13 ^#^**	**60.86 ± 12.37 ^#^**
Sex (Male/Female) *n*	4/13	35/53	23/17	**13/35 ^#^**	**1/19 ^#^**	**12/16 ^&^**
GAP stage *n* (%)						
Stage I	NA	65 (74.7)	23 (59.0)	42 (87.5)	18 (90.0)	24 (85.7)
Stage II	NA	19 (21.8)	13 (33.3)	6 (12.5)	2 (10.0)	4 (14.3)
Stage III	NA	3 (3.5)	3 (7.69)	0	0	0
HRCT pattern *n* (%)						
NSIP *n* (%)	NA	9 (10.2)	0	9 (18.7)	**8 (40.0) ^#^**	**1 (3.6) ^&^**
UIP *n* (%)	NA	27 (30.7)	24 (60.0)	3 (6.3)	**2 (10.0) ^#^**	**1 (3.6) ^#^**
pUIP *n* (%)	NA	27 (30.7)	16 (40.0)	11 (22.9)	4 (20.0)	7 (25.0)
Other *n* (%)	NA	25 (28.4)	0	25 (52.1)	**6 (30.0) ^#^**	**19 (67.8) ^#^**
Laboratory parameters *n* (%)						
ANA	NA	43 (48.9)	23 (57.5)	20 (41.7)	8 (40.0)	12 (42.9)
RF	NA	19 (21.6)	6 (15.0)	13 (27.1)	6 (30.0)	7 (25.0)
ACCP	NA	2 (2.3)	1 (2.5)	1 (2.1)	1 (5.0)	0
Anti-cytoplasmatic	NA	21 (23.9)	8 (20.0)	13 (27.1)	7 (35.0)	6 (21.4)
Anti-chromatin	NA	29 (33.0)	15 (37.5)	14 (29.2)	7 (35.0)	7 (25.0)
Anti-Jo-1	NA	0	0	0	0	0
Anti-SSA	NA	7 (8.0)	2 (5.0)	5 (10.4)	4 (20.0)	1 (3.6)
Anti-SSB	NA	1 (1.1)	0	1 (2.1)	1 (5.0)	0
Anti-SCL-70	NA	3 (3.4)	0	3 (6.3)	3 (15.0)	0
Anti-RNP	NA	1 (1.1)	0	1 (2.1)	1 (5.0)	0
Anti-dsDNA	NA	NA	NA	3 (6.3)	3 (15.0)	0
ANCA	NA	13 (14.8)	8 (20.0)	5 (10.4)	1 (5.0)	4 (14.3)
CRP (mg/L)	NA	6.53 (2.47–12.79) (*n* = 78)	6.82 (2.42–12.44) (*n* = 33)	6.24 (2.31–13.21) (*n* = 45)	8.83 (3.13–15.51) (*n* = 19)	5.91 (2.09–11.26) (*n* = 26)
eGFR (mL/min/1.73 m^2^)	NA	86.15 (71.35–90.00)	88.05 (76.88–90.00)	82.60 (58.40–90.00)	90.00 (74.20–90.00)	78.70 (57.80–90.00)

ACCP, anti-cyclic citrullinated peptide antibodies; ANA, antinuclear antibodies; ANCA, anti-neutrophil cytoplasmic antibodies; anti-dsDNA, antibodies to double-stranded deoxyribonucleic acid; anti-SCL-70, anti-topoisomerase I antibodies; anti-SSA, Ro autoantibodies; anti-SSB, anti-La antibodies; anti-RNP, antibodies to ribonucleoprotein; CRP, C-reactive protein; eGFR, estimated glomerular filtration rate; GAP, gender–age–physiology index; HRCT, high-resolution computed tomography; NSIP, non-specific interstitial pneumonia; pUIP, probable usual interstitial pneumonia; RF, rheumatoid factor; UIP, usual interstitial pneumonia. For better readability, only the significant results were marked in bold in the table. In the posthoc analyses of Chi Square-tests, statistical significance was determined at *p* < 0.0167 for sex; and at *p* < 0.0042 for HRCT pattern. * significant difference vs. control, ^#^ significant difference vs. IPF; ^&^ significant difference vs. CTD-ILD.

**Table 2 ijms-24-09178-t002:** Lung function, ABG, 6MWT functional parameters.

Parameters	Healthy Controls (*n* = 17)	All ILD Patients (*n* = 88)	IPF (*n* = 40)	Non-IPF ILD (*n* = 48)	CTD-ILD (*n* = 20)	Other ILD (*n* = 28)
Lung function						
FVC (L)	NA	2.33 (1.72–3.24)	2.35 ± 0.82	2.62 ± 1.10	2.46 ± 0.97	2.72 ± 1.20
FVC (% predicted)	NA	77.72 ± 25.08	74.5 (55.25–82.25)	81.08 ± 26.38	79.45 ± 29.72	80.35 ± 22.34
FEV_1_ (L)	NA	1.94 (1.52–2.60)	1.95 ± 0.55	2.03 (1.48–2.81)	2.08 ± 0.78	1.99 (1.48–2.82)
FEV_1_ (% predicted)	NA	78.22 ± 22.52	77 (60.00–86.25)	79.43 ± 25.15	79.45 ± 29.72	77.80 ± 22.96
FEV_1_/FVC (%)	NA	84.26 (78.11–90.12)	85.79 (77.66–91.95)	83.73 (78.15–88.77)	85.46 ± 7.19	81.53 (77.92–88.43)
TLC (L)	NA	4.30 ± 1.45 (*n* = 79)	4.25 ± 1.38 (*n* = 36)	4.34 ± 1.51 (*n* = 43)	3.96 ± 1.095 (*n* = 17)	4.59 ± 1.70 (*n* = 26)
TLC (%predicted)	NA	75.71 ± 22.01 (*n* = 79)	68.50 (59.00–85.75) (*n* = 36)	76.81 ± 21.84 (*n* = 43)	74.12 ± 19.12 (*n* = 17)	77.84 ± 22.91 (*n* = 26)
Diffusion parameters						
T_LCO_ (mmol/min/kPa)	NA	5.38 (4.33–7.43) (*n* = 87)	4.96 ± 1.56 (*n* = 39)	**6.49 ± 2.47 ****	6.18 ± 2.18	**6.71 ± 2.68 ****
T_LCO_ (%predicted)	NA	69.00 (56.00–85.00) (*n* = 87)	60.92 ± 17.10 (*n* = 39)	**76.40 ± 23.05 ***	**73.20 ± 20.59 ***	**78.39 ± 24.95 ****
K_LCO_ (mmol/min/kPa/L)	NA	1.32 ± 0.44 (*n* = 87)	1.16 ± 0.38 (*n* = 39)	**1.46 ± 0.44 ****	**1.53 ± 0.47 ****	**1.41 ± 0.41 ***
K_LCO_ (%predicted)	NA	70.18 ± 21.60 (*n* = 87)	64.10 ± 19.22 (*n* = 39)	**75.13 ± 22.35 ***	75.65 ± 21.76	75.07 ± 22.87
ABG						
pH	NA	7.419 (7.403–7.432) (*n* = 76)	7.414 (7.395–7.430) (*n* = 38)	**7.421 (7.412–7.436) *** **(*n* = 38)**	7.422 (7.407–7.439) (*n* = 16)	7.424 ± 0.020 (*n* = 22)
pCO_2_ (mmHg)	NA	38.55 (35.60–40.95) (*n* = 80)	38.95 (37.00–42.58)	**36.90 (34.93–40.38) *** **(*n* = 40)**	37.00 (35.50–40.95) (*n* = 17)	37.26 ± 3.28 (*n* = 23)
pO_2_ (mmHg)	NA	66.99 ± 11.19 (*n* = 80)	63.92 ± 10.17	**70.06 ± 11.43 *** **(*n* = 40)**	**72.27 ± 8.96 *** **(*n* = 17)**	67.91 ± 13.25 (*n* = 23)
6MWT						
Distance (m)	NA	388.58 ±142.72 (*n* = 52)	386.56 ± 126.39 (*n* = 32)	391.80 ± 169.10 (*n* = 20)	416.14 ± 162.06 (*n* = 7)	378.69 ± 177.79 (*n* = 13)
SpO_2_ baseline (%)	NA	96.00 (92.00–97.00) (*n* = 47)	96.00 (92.00–97.00) (*n* = 31)	93.69 ± 4.84 (*n* = 16)	95.80 ± 3.27 (*n* = 5)	92.73 ± 5.26 (*n* = 11)
SpO_2_ post-exercise (%)	NA	86.00 (79.00–91.00) (*n* = 47)	84.87 ± 7.72 (*n* = 31)	83.06 ± 11.39 (*n* = 16)	84.2 ± 11.08 (*n* = 5)	82.55 ± 12.02 (*n* = 11)
Desaturation (%)	NA	9.79 ± 6.98 (*n* = 47)	9.36 ± 6.12 (*n* = 31)	10.63 ± 8.56 (*n* = 16)	11.6 ± 8.33 (*n* = 5)	10.18 ± 9.03 (*n* = 11)
Pulse baseline (1/min)	NA	78.73 ± 14.74 (*n* = 45)	77.10 ± 14.07 (*n* = 30)	82.00 ± 16.00 (*n* = 15)	66.67 ± 32.48 (*n* = 6)	84.10 ± 14.50 (*n* = 10)
Pulse post-exercise (1/min)	NA	108.10 ± 21.31 (*n* = 44)	108.20 ± 20.27 (*n* = 29)	107.90 ± 23.93 (*n* = 15)	102.00 ± 32.33 (*n* = 5)	110.80 ± 19.94 (*n* = 10)
Borg scale baseline (0–10)	NA	0 (1–0.625) (*n* = 46)	0 (0–1) (*n* = 30)	0 (0–0.375) (*n* = 16)	0 (0–0) (*n* = 5)	0 (0–1) (*n* = 11)
Borg scale post-exercise (0–10)	NA	3 (1–5) (*n* = 47)	4 (2–5) (*n* = 31)	3 (1–4) (*n* = 16)	3 (1.25–4.5) (*n* = 5)	3 (1–4) (*n* = 11)

6MWT, 6-min walk test; ABG, arterialized capillary blood gases; FVC, forced vital capacity; FEV_1_, forced expiratory volume in 1 s; TLC, total lung capacity; T_LCO_, transfer factor of the lung for carbon monoxide; K_LCO_, transfer coefficient of the lung for carbon monoxide; pO_2_, partial pressure of oxygen; pCO_2_, partial pressure of carbon dioxide; SpO_2_, saturation of peripherial oxygen. Significant results are marked in bold in the table. * *p* < 0.05 vs. IPF, ** *p* < 0.01 vs. IPF.

## Data Availability

The data presented in this study are available on request from the corresponding author.

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
