# Peer review of "Serum Progranulin Level Might Differentiate Non-IPF ILD from IPF"

_ijms, 2023, doi:10.3390/ijms24119178_

Round 1

Reviewer 1 Report

Except for the relative low number and the therefore biased NON-IPF-ILD group the idea to differentiate along a biomarker is highly needed. Go ahead !

Author Response

Dear Reviewer 1,

enclosed please find the revised version of our manuscript entitled "Serum progranulin level might differentiate non-IPF ILD from IPF" to publish in International Journal of Molecular Sciences.

We are grateful for the effort of the Reviewers and the Editorial Office, and we hope you find our responses satisfactory.

Please find our response to your comment below.

Yours sincerely, Authors

Comment 1: Except for the relative low number and the therefore biased NON-IPF-ILD group the idea to differentiate along a biomarker is highly needed. Go ahead !

Response 1: We would like to thank the Reviewer for the valuable feedback. We entirely agree that the low numbers are a limitation of our study as we indicated in the Discussion.

Reviewer 2 Report

Comments to the Author

The authors suggested that progranulin can differentiate IPF with non-IPF-ILD.

This study has clinical significance and novelty. The resolution of the following issues for acceptance of International Journal of Molecular Science.

Major comment

1.     Method: The author should show that the diagnoses with idiopathic interstitial pneumonia and each connective tissue disease were based on global guidelines. In the diagnoses of ILD subtype in study, central diagnoses by multidisciplinary discussion diagnoses and radiological diagnoses, in particular differing UIP pattern from non-UIP pattern by more than two radiological experts are required. The author should reveal whether those diagnostic methods were performed. If not, the present study has study limitation.

2.     Method and Result: The author should explain when sera were collected (e.g. at diagnoses of ILD or before start of drug therapy). If time of collecting sera was heterogeneous, the author should reveal duration from diagnoses of ILD to collecting sera.

3.     Result: Do the differences of drug therapy, in particular steroid, immunosuppressants influence serum progranulin level? The author showed that many patients with IPF was treated with antifibrotic patients at the time of collecting sera in Discussion section but not clarified rate of patients treated with steroid and immunosuppressants. I concerned that use rate of steroid and/or immunosuppressant was higher in non-IPF patients comparing with IPF patients and influenced the results. The author should show details of treatment for study subjects at the time of collecting sera and discuss whether those use influence the result of the present study.

4.     Result: In the comparison of serum progranulin level between IPF and non-IPF, and between UIP and non-UIP, please show the results of a two-group comparison after adjusting for some confounding factors that affect the difference between IPF and non-IPF.

Author Response

Dear Reviewer 2,

enclosed please find the revised version of our manuscript entitled "Serum progranulin level might differentiate non-IPF ILD from IPF" to publish in International Journal of Molecular Sciences.

We are grateful for the effort of the Reviewers and the Editorial Office, and we hope you find our responses satisfactory.

Please find our point-by-point responses to the comments below.

Yours sincerely, Authors

Comment 1: Method: The author should show that the diagnoses with idiopathic interstitial pneumonia and each connective tissue disease were based on global guidelines. In the diagnoses of ILD subtype in study, central diagnoses by multidisciplinary discussion diagnoses and radiological diagnoses, in particular differing UIP pattern from non-UIP pattern by more than two radiological experts are required. The author should reveal whether those diagnostic methods were performed. If not, the present study has study limitation.

Response 1: We entirely agree with the Reviewer, the ILD subgroup was identified for each patient by the multidisciplinary team of our ILD center according to the international guidelines. We added the following lines to the Methods for clarification: “Our MDD team consists of 3 pulmonologists, expert radiologists, clinical immunologists with expertise in ILD and pathologist if histology present, according to international guidelines [14, 15].”

Comment 2: Method and Result: The author should explain when sera were collected (e.g. at diagnoses of ILD or before start of drug therapy). If time of collecting sera was heterogeneous, the author should reveal duration from diagnoses of ILD to collecting sera.

Response 2: The Reviewer raised an important point. As this was a cross-sectional analysis with prevalent patients, differences in therapy might have influenced the results. Patients had their sample taken when coming to a regular check-up, so this sampling methodology represents real world practice. Methods were supplemented with the following: “All patients were diagnosed with ILD within 2 months to 3 years prior to the sampling and most of them were on stable dose antifibrotic or immunosupressive medication for at least 4 weeks at the time of the testing.”

Comment 3: Result: Do the differences of drug therapy, in particular steroid, immunosuppressants influence serum progranulin level? The author showed that many patients with IPF was treated with antifibrotic patients at the time of collecting sera in Discussion section but not clarified rate of patients treated with steroid and immunosuppressants. I concerned that use rate of steroid and/or immunosuppressant was higher in non-IPF patients comparing with IPF patients and influenced the results. The author should show details of treatment for study subjects at the time of collecting sera and discuss whether those use influence the result of the present study.

Response 3: We completely agree with the Reviewer, differences in medication between the ILD groups should be considered. There are literature data that suggest potential effect of immunosuppressive therapy on PGRN levels. We took into consideration the applied immunosuppressive and antifibrotic treatment in the performed linear regression analysis, which is discussed in detail in Response 4. The following statement was included into the Discussion: “There are limited data available about potential influence of immunosupressants, especially steroids on the level of PGRN [30]. Our results showed that neither immunosuppressive, nor antifibrotic treatment was associated with the PGRN levels.”

Comment 4: Result: In the comparison of serum progranulin level between IPF and non-IPF, and between UIP and non-UIP, please show the results of a two-group comparison after adjusting for some confounding factors that affect the difference between IPF and non-IPF.

Response 4: The Reviewer raised an important point, that potential confounding factors were not considered in the assessment of differences in PGRN levels between IPF and non-IPF ILD, and UIP and non-UIP. We performed a multivariate linear regression analysis including age, sex and therapy and added the following sections to the manuscript:

Methods: “After performing the univariate tests, in order to assess the role of possible confounding factors on the PGRN levels, we performed a stepwise multivariate linear regression analysis on the univariate data that proved to be significant according to the following steps: Step 1. age and sex, Step 2. medication included, Step 3. presence of ILD included, Step 4. IPF – non-IPF diagnosis included, Step 5. HRCT pattern included (Table 1. in Supplementary Material).”

Results: “Five-step multivariate linear regression analysis showed that higher age and male sex predicts higher PGRN levels, and confirmed that IPF diagnosis associates with lower PRGN levels than non-IPF, however, based on step 5 we can assume that UIP pattern is more strongly associated with lower PGRN levels than an IPF diagnosis. Immunosuppressive and antifibrotic treatment did not influence the PGRN values (Table 1. in Supplementary Material).”

Discussion: “There are limited data available about potential influence of immunosupressants, especially steroids on the level of PGRN [30]. Our results showed that neither immunosuppressive, nor antifibrotic treatment was associated with the PGRN levels. Our linear regression model also indicated that higher age and male sex predicts higher levels of PGRN. Although the IPF group was significantly older and more prominently male, linear regression still showed strong association between IPF and lower PGRN levels.”

Stepwise linear regression results have been added as supplementary material.

Reviewer 3 Report

Interstitial lung diseases (ILD) diagnosis is an intriguing process. Biomarkers elucidation may be very supportive for diagnosis. Elevated serum progranulin (PGRN) levels have been reported in liver fibrosis and dermatomyositis-associated acute interstitial pneumonia. The authors attempt to assess the role of PGRN in the differential diagnosis of idiopathic pulmonary fibrosis and other ILDs is interesting.

Major Comments

1.    The serum PGRN levels are significantly correlated with serum IL-6 levels. It will be a good idea to analyze the serum IL-6 levels among healthy, all patients, IPF and non-IPF-ILD groups.

2.    Please include diagnostic factors for PF-ILD by employing univariate logistic regression analysis to assess the predictive markers. It will be interesting to see the impact of gender, age, higher BALF NLR, BALF neutrophil percentage, lower BALF lymphocyte percentage, BALF eosinophil percentage and CD8 levels.

Minor Comments

1.    Please provide graphical abstract of the complete study for ease in understanding.

Author Response

Dear Reviewer 3,

enclosed please find the revised version of our manuscript entitled "Serum progranulin level might differentiate non-IPF ILD from IPF" to publish in International Journal of Molecular Sciences.

We are grateful for the effort of the Reviewers and the Editorial Office, and we hope you find our responses satisfactory.

Please find our point-by-point responses to the comments below.

Yours sincerely, Authors

Major Comments

Comment 1:    The serum PGRN levels are significantly correlated with serum IL-6 levels. It will be a good idea to analyze the serum IL-6 levels among healthy, all patients, IPF and non-IPF-ILD groups.

Response 1: We fully agree with the Reviewer, that IL-6 levels can influence the progranulin levels as stated in the literature, however, unfortunately we did not measure IL-6. We added the following sentence into the Discussion: “There are potential factors such as IL-6 [11,30] and TNF-α [30] that can influence the serum PGRN levels, however in this study these cytokine levels were not measured” and the new reference was included.

Comment 2:    Please include diagnostic factors for PF-ILD by employing univariate logistic regression analysis to assess the predictive markers. It will be interesting to see the impact of gender, age, higher BALF NLR, BALF neutrophil percentage, lower BALF lymphocyte percentage, BALF eosinophil percentage and CD8 levels.

Response 2: We would like to thank the Reviewer for highlighting the very important aspect that potential confounding factors were not considered in the assessment of differences in PGRN levels between IPF and non-IPF ILD. We performed a multivariate linear regression analysis including age, sex and therapy and added the following sections to the manuscript:

Methods: “After performing the univariate tests, in order to assess the role of possible confounding factors on the PGRN levels, we performed a stepwise multivariate linear regression analysis on the univariate data that proved to be significant according to the following steps: Step 1. age and sex, Step 2. medication included, Step 3. presence of ILD included, Step 4. IPF – non-IPF diagnosis included, Step 5. HRCT pattern included (Table 1. in Supplementary Material).”

Results: “Five-step multivariate linear regression analysis showed that higher age and male sex predicts higher PGRN levels, and confirmed that IPF diagnosis associates with lower PRGN levels than non-IPF, however, based on step 5 we can assume that UIP pattern is more strongly associated with lower PGRN levels than an IPF diagnosis. Immunosuppressive and antifibrotic treatment did not influence the PGRN values (Table 1. in Supplementary Material).”

Discussion: “There are limited data available about potential influence of immunosupressants, especially steroids on the level of PGRN [30]. Our results showed that neither immunosuppressive, nor antifibrotic treatment was associated with the PGRN levels. Our linear regression model also indicated that higher age and male sex predicts higher levels of PGRN. Although the IPF group was significantly older and more prominently male, linear regression still showed strong association between IPF and lower PGRN levels.”

Stepwise linear regression results have been added as supplementary material.

Unfortunately, we did not have sufficient data on the patient's BALF cell profiles to be able to analyze its effect on the PGRN level.

Minor Comments

Comment 1:    Please provide graphical abstract of the complete study for ease in understanding.

Response 1: We would like to thank the Reviewer for the helpful suggestion, we added a graphical abstract to the manuscript.

Round 2

Reviewer 2 Report

Author correctly responded to comments.